# The role of race and insurance in trauma patients' mortality: A cross-sectional analysis based on a nationwide sample

**José A. Acosta**[ID]*

New Mexico Department of Health, Santa Fe, New Mexico, United States of America

* jose.acostasd@gmail.com

## Abstract

### Background

Persistent disparities in trauma in-hospital mortality owing to insurance status and race remain a prominent issue within healthcare. This study explores the relationships among insurance status, race, length of stay (LOS) in-hospital mortality outcomes in trauma patients at extreme risk of mortality (EROM) trauma patients.

### Methods

Data was retrieved from the National Inpatient Sample, focusing on high-acuity trauma patients from 2007 to 2020, aged 18–64 years. Patients were identified using specific All Patient Refined Diagnosis Related Groups codes. Emphasis was placed on those with EROM owing to their resource-intensive nature and the potential influence of insurance on outcomes. Patients aged 65 years or older were excluded owing to distinct trauma patterns, as were those diagnosed with burns or non-trauma conditions.

### Results

The study encompassed 70,381 trauma inpatients with EROM, representing a national estimate of 346,659. Being insured was associated with a 34% decrease in the odds of in-hospital mortality compared to being uninsured. The in-hospital mortality risk associated with insurance status varied over time, with insurance having no impact on in-hospital mortality during hospitalizations of less than 2 days (short LOS). In the overall group, Black patients showed an 8% lower risk of in-hospital mortality compared to White patients, while they experienced a 33% higher risk of in-hospital mortality during short LOS.

### Conclusion

Insured trauma inpatients demonstrated a significant reduction in the odds of in-hospital mortality compared to their uninsured counterparts, although this advantage was not present in the short LOS group. Black patients experienced lower in-hospital mortality rates compared to White patients, but this trend reversed in the short LOS group. These findings

**Data Availability Statement:** The data underlying the results presented in the study are available from Healthcare Cost and Utilization Project at https://hcup-us.ahrq.gov/.

**Funding:** The author(s) received no specific funding for this work.

**Competing interests:** The authors have declared that no competing interests exist.

underscore the intricate relationships between insurance status, race, and duration of hospitalization, highlighting the need for interventions to improve patient outcomes.

## Introduction

Trauma stands as a formidable challenge in global public health, consistently ranking as a leading cause of mortality and prolonged disability, particularly among the youth and young adult population. The intricacies of patient outcomes following traumatic injuries are influenced by a myriad of factors, ranging from the availability of medical resources and the proficiency of the medical team to broader socio-economic determinants, among which insurance status is a pivotal factor [1].

Examining the existing body of literature unveils a landscape filled with varied interpretations and conclusions regarding the nexus between insurance coverage, racial demographics, and the outcomes of trauma patients [2–8]. In the realm of trauma in-hospital mortality, the contrast between the insured and uninsured has generated varying conclusions within the existing literature, based on the timing and status of insurance coverage [4,9–18]. Most reports in the literature highlight racial disparities regarding trauma in-hospital mortality, especially among Black trauma patients; however, some reports dispute these findings, indicating a divergence of perspectives on this issue [4,19–21]. The discordance in these findings underscores the complexity of the issue and highlights a significant gap in our understanding of the underlying factors at play.

Another prominent gap in our current understanding of the effect of insurance and race on in-hospital trauma mortality arises from the biases that are intrinsic to the data and methodologies employed in previous studies. Time-dependence bias is pivotal factor that can skew interpretations and lead to conflicting conclusions [4,22,23]. This bias becomes evident when patients with extended duration of hospitalizations are more likely to obtain insurance, potentially resulting in improved outcomes. Several studies investigating insurance's impact on trauma mortality lack stratification for duration of hospitalization, potentially introducing time-dependent bias concerning insurance's effect [9,14,18].

Furthermore, the enactment of the Emergency Medical Treatment and Labor Act (EMTALA) in 1986 to address the unethical practice of "patient dumping" may potentially contribute to time-dependence bias when examining in-hospital trauma mortality. By mandating that emergency departments provide critical care to all patients, regardless of their insurance status or financial means, EMTALA aimed to establish equal access to emergency medical services. However, it is essential to recognize that while this act ensures immediate access to medical attention, it does not guarantee uniform quality or continuity of care in the post-stabilization phase [9,15,24–26].

This study endeavors to bridge crucial gaps in the literature, specifically concerning the divergent impacts of insurance, race, and time-dependent effects on trauma in-hospital mortality. It does so by utilizing a stratified model to analyze a large publicly available database, National Inpatient Sample (NIS), with a particular focus on trauma inpatients at an extreme risk of mortality (EROM) [27]. The primary aim is to offer specific and actionable insights that can inform evidence-based policy interventions.

## Methods

A retrospective cross-sectional analysis was employed, using data from the National Inpatient Sample (NIS) database managed by the Agency for Healthcare Research and Quality, spanning

2007–2020. Its systematic data collection methodology, involving a representative sample of hospitals across the country and careful weighting, allows for the reliable extrapolation of findings to national estimates. The annual NIS datasets, derived from 7–8 million individual hospitalizations across non-federal, short-term, general, and specialized community hospitals in 48 US states and Washington DC, yield a national estimate of 35–39 million hospitalizations [27,28].

In the NIS, each record represents a hospitalization event, not a unique patient [28]. For this study, to simplify statistical analysis and interpretation, each hospitalization is considered and referred to as a separate patient encounter. This approach is applied even in cases where individuals may have more than one admission for the same clinical issue.

The analysis targeted trauma patients aged 18–64 years, identified using All Patient Refined Diagnosis Related Groups (APRDRG) codes: 20, 40, 55–57, 135, 308, 384, 910–912, and 930. Patients aged 65 years or older were excluded because their general patterns of trauma differ from those of the younger population. The APRDRG system, developed by 3M™ Health Information Systems, stratifies based on patient severity and risk. The study concentrated on patients categorized at an EROM, excluding those with primary burn or non-trauma diagnoses [29]. Patients in the EROM group were chosen for this study due to their higher resource-intensive care needs, which may be influenced by their insurance status. During the study period, the APRDRG versions underwent changes, starting with version 25.1 in 2007 and advancing to version 38.0 in 2020.

The descriptive data and statistical analysis of the NIS accounted for its complex survey design, specifically addressing stratification, clustering, and weighting which allows for national estimates. The study employed "survey data analysis modules" available in statistical programs to ensure precise accounting for complex survey design characteristics during statistical evaluations.

Insurance status served as the primary exposure in the investigation of in-hospital mortality among EROM trauma patients. For insurance classifications, self-payers were labeled uninsured, while those with absent insurance details were categorized as having missing data. Patients under Medicare, Medicaid, or private policies were considered insured. Medicare, traditionally for those aged 65 years or older, covers a subset below this age owing to specific conditions such as end-stage renal disease.

Assessing the impact of duration of hospitalization on in-hospital mortality involved categorizing duration into two groups based on length of stay (LOS): short LOS, comprising stays less than two days; and long LOS, encompassing stays of two days or more. A multivariate logistic regression model was employed, adjusting for age, race, sex, median household income and Affordable Care Act Medicaid Expansion (ACAME) enactment in 2014. The expansion of Medicaid provides health insurance to those under 65 with incomes up to 133% of the US federal poverty level [30]. In the sensitivity analysis, an interaction term was incorporated into the multivariate logistic regression model to evaluate if insurance status influenced the relationship between race and outcomes.

The "missing at random" principle was applied to handle missing data, considering the nature and collection methodology of the NIS data. Specifically, the race variable presented with 10.07% missing data, while all other variables had less than 5% missing. To address missing race data, multiple imputation via chained equations was employed, utilizing multinomial logistic regression analysis to accommodate the categorical nature of the race variable.

The data provided by NIS is de-identified and fully anonymized to ensure the privacy and confidentiality of individual patients. Therefore, there are no directly identifiable patient details within the dataset. The Solutions IRB, LLC, accredited by the Association for the Accreditation of Human Research Protection Programs, classified this study as non-human

subject research upon evaluation. This fully anonymized nature of the dataset ensures that the study met the criteria for exemption from informed consent. The reference to 3M™ Health Information Systems, a commercial product, is made impartially, without any conflict of interest.

The manuscript was prepared according to the methodology stipulated by the EQUATOR Network reporting using the Strengthening the Reporting of Observational Studies in Epidemiology (STROBE) checklist found in the supporting information section. Stata version 18 MP, incorporating the survey data analysis (svy) and multiple imputation (mi) modules, served as the software for statistical analysis, with significance set at p < 0.05. Furthermore, Wordtune Editor and ChatGPT were employed to refine grammar and clarity of the manuscript. [31,32].

## Results

From January 2007 to December 2020, the NIS captured a total of 103,434,123 hospitalizations, which, under the assumption of equating hospitalizations with individual patients represents an estimated 506,403,411 individuals. Among those individuals aged 18–64 years, 715,368 met the criteria for trauma, corresponding to a national estimate of 3,512,361. Of these, 70,381 had EROM, which correlated with a national estimate of 346,659, forming the basis for this study (Fig 1).

The descriptive data were based on national estimates. In the dataset, 82,179 patients (23.7%) had a short LOS, and 264,480 (76.3%) patients had a long LOS. Descriptive data are presented in Table 1.

Concerning sex distribution, both groups had a higher proportion of men (77.6% in short LOS and 75.9% long LOS) compared to women. The racial distribution was consistent across both groups, with the majority identifying as White (61.6% in short LOS and 61.2% in long LOS), followed by Blacks and Hispanics. Regarding socio-economic status, as measured by median household income quartile, a greater proportion of patients in both LOS groups fell into the lowest income quartile (36.1% in short LOS and 35.5% in long LOS). In the short LOS group, in-hospital mortality was 74.3%, while in the long LOS group, was 17.4%. Insurance status showed that a substantial majority of patients in both groups are insured, with 74.8% in the short LOS group and an even higher 86.2% in the long LOS group.

Table 2 presents the results of the multivariate logistic regression analysis. Insured patients saw a 34% decrease on in-hospital mortality odds (OR = 0.66, 95% CI: 0.62–0.69; p = < 0.001) relative to the uninsured. However, the influence of insurance varied by LOS. For patients with short LOS, insurance status did not significantly affect in-hospital mortality (OR = 0.93, 95% CI: 0.83–1.05; p = 0.24); however, insured patients with long LOS experienced a 17% decrease in mortality risk (OR = 0.83, 95% CI: 0.77–0.89; p = <0.001).

Overall, compared to White patients, Black patients experienced a reduction in in-hospitals mortality odds by 8% (OR = 0.92, 95% CI: 0.87–0.96; p = 0.001). Moreover, the overall in-hospital mortality rates for Hispanics, Asian/Pacific Islanders, and Native Americans showed no significant differences compared to the reference group. In the short LOS group, Black patients showed an 33% increase (OR = 1.33, 95% CI: 1.13–1.57; p = 0.001) in-hospital mortality and Native Americans showed a significant 40% reduction in-hospital mortality odds compared to the reference group (OR = 0.60, 95% CI: 0.38–0.94; p = 0.03). In the group with long LOS, Blacks patients experienced a significant 12% reduction in the odds of in-hospital mortality (OR = 0.88, 95% CI: 0.83–0.94; p = <0.001), whereas Hispanics Asian/Pacific Islander had a substantial increase in these odds of 9% (OR = 1.09, 95% CI: 1.02–1.18; p = 0.02) and 19% (OR = 1.19, 95% CI: 1.01–1.41: p = 0.04)respectively.

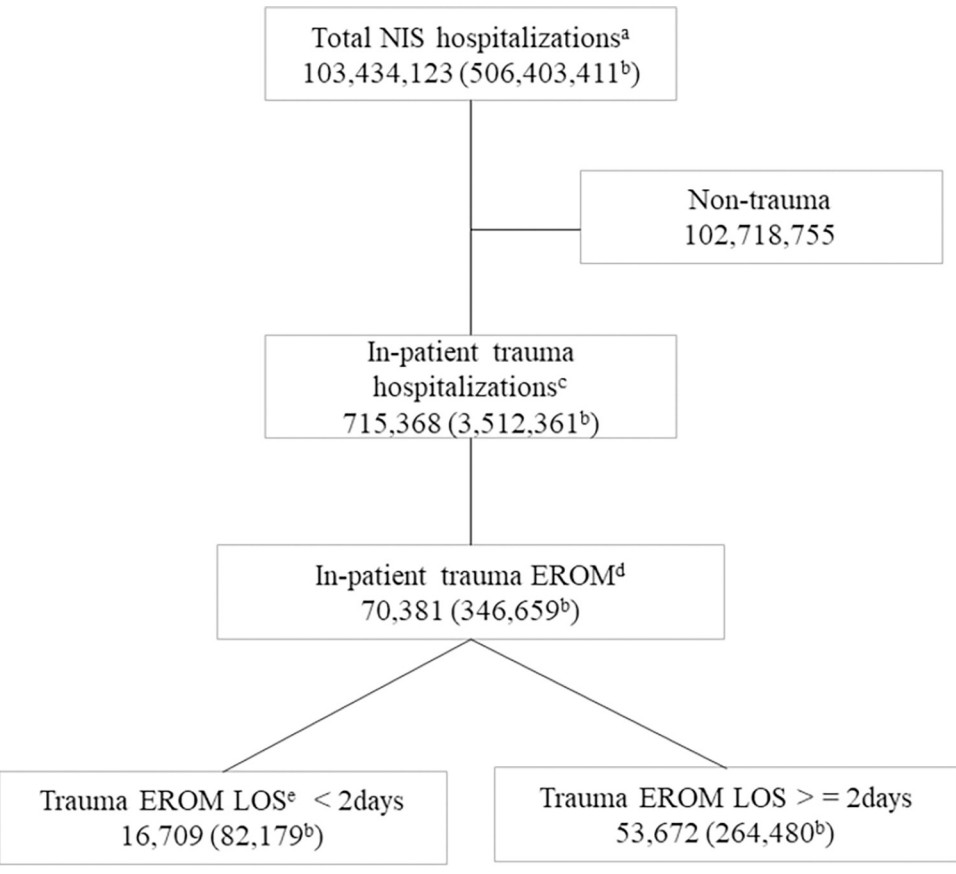

a. Hospitalizations are treated as a distinct patient encounter (see methods section for details)
b. ( ) = National estimates
c. 18-64 years of age
d. EROM = Extreme risk of mortality; 18-64 years of age
e. LOS = length of stay

**Fig 1. National Inpatient Sample patient flow diagram.**

Upon examining the impact of median household income on in-hospital mortality, overall patients in the 76–100 income percentile had a 7% (OR = 0.93, 95% CI: 0.88–0.99; p = 0.02 reduction in in-hospital mortality odds compared to the reference group (0-25th percentile). Overall, in-hospital mortality odds were reduced by 27% (OR = 0.73, 95% CI: 0.70–0.77; p = <0.001) in trauma patients at EROM when admitted after the enactment of ACAME in 2014.

The sensitivity analysis incorporated an interaction term between insurance and race to examine if insurance status influenced outcomes differently across racial groups. The lack of statistical significance for the interaction term except for the interaction term between insurance and Black patients in the long LOS (OR = 1.19; 95% CI: 1.01–1.39; p = 0.03) and insurance and Hispanic patients in the overall group (OR = 1.16, 95% CI: 1.02–1.32; p = 0.03) suggests that there may not be a strong interactive effect between insurance and race on the likelihood of in-hospital mortality (S1 Appendix).

**Table 1. National estimates of descriptive trauma patients at extreme risk of mortality 2007–2020.**

|  | Short LOS[a] | Long LOS[b] |
|---|---|---|
| N[c] | 82,179 (23.7%) | 264,480 (76.3%) |
| Age (mean SD) | 39.07 (14.42) | 41.90 (14.37) |
| Sex |  |  |
| Male | 63,657 (77.6%) | 200,726 (75.9%) |
| Female | 18,331 (22.4%) | 63,636 (24.1%) |
| Race |  |  |
| White | 43,683 (61.6%) | 147,555 (61.2%) |
| Black | 12,186 (17.2%) | 43,882 (18.2%) |
| Hispanic | 9,196 (13.0%) | 32,834 (13.6%) |
| Asian/Pacific Islander | 1,187 (1.7%) | 4,101 (1.7%) |
| Native American | 1,046 (1.5%) | 3,127 (1.3%) |
| Other | 3,621 (5.1%) | 9,652 (4.0%) |
| Median household income quartile |  |  |
| 0–25th | 28,246 (36.1%) | 90,410 (35.5%) |
| 26 to 50th | 20,430 (26.1%) | 67,312 (26.4%) |
| 51 to 75th | 17,546 (22.4%) | 56,985 (22.4%) |
| 76 to 100th | 12,043 (15.4%) | 39,809 (15.6%) |
| In-hospital mortality | 61,016 (74.3%) | 45,992 (17.4%) |
| Insurance |  |  |
| No | 20,292 (25.2%) | 35,970 (13.8%) |
| Yes | 60,191 (74.8%) | 224,872 (86.2%) |

[a] Short LOS = length of stay < 2 days.

[b] Long LOS = length of stay ≥ 2 days.

[c] N = National estimate of the number of trauma patients at an extreme risk of mortality.

## Discussion

This study examines the associations between insurance status, race, LOS, and in-hospital mortality of trauma patients at EROM. Overall, insured patients exhibited a 34% reduced likelihood of in-hospital mortality. However, when assessing LOS, the relationship becomes complex. Patients in the short LOS group did not reveal a significant link between insurance and mortality. Yet, for patients in the long LOS group, insurance was associated with a 17% decrease of in-hospital mortality odds. Such patterns indicate the potential for heterogeneity in outcomes based on time since admission. Early fatalities in trauma often arise from immediate consequences such as hemorrhagic shock or severe brain injuries, while deaths occurring later might result from complications like sepsis or multi-organ failure [33–35].

Another explanation for this difference in outcomes related to the insurance status could be EMTALA, which mandates hospitals to provide care in emergencies irrespective of patients' insurance status. While EMTALA primary goals is to prevent "patient dumping" its impact on patient outcomes is more indirect. It doesn't guarantee a specific quality of care beyond the initial screening and stabilization [24]. Patient outcomes depend on various factors, including the severity of the medical condition, the quality of care provided, and the patient's overall health.

The findings related to insurance in this analysis significantly deviate from prior reports. The contrasting outcomes stem from the varied data sources employed in different research projects. Past investigations have relied on the American College of Surgeons National Trauma Data Bank (NTDB) [2,5–9,11–13,18,36]. However, this database, while substantial, might not

**Table 2. Logistic regression analysis of in-hospital mortality for trauma patients at extreme risk of mortality[a].**

|  | Short LOS[b] | | | Long LOS[c] | | | Overall | | |
|---|---|---|---|---|---|---|---|---|---|
|  | Odds ratio | 95% confidence interval | p | Odds ratio | 95% confidence interval | p | Odd ratio | 95% confidence interval | p |
| Insurance | 0.93 | 0.83 1.05 | 0.24 | 0.83 | 0.77 0.89 | < 0.001 | 0.66 | 0.62 0.69 | < 0.001 |
| Age | 1.01 | 1.01 1.02 | < 0.001 | 1.01 | 1.01 1.01 | < 0.001 | 1.00 | 1.00 1.01 | <0.001 |
| Sex | 1.20 | 1.07 1.35 | 0.002 | 0.94 | 0.90 0.99 | 0.02 | 0.98 | 0.94 1.02 | 0.27 |
| Race |  | | |  | | |  | | |
| White | Reference | | |  | | |  | | |
| Black | 1.33 | 1.13 1.57 | 0.001 | 0.88 | 0.83 0.94 | <0.001 | 0.92 | 0.87 0.96 | 0.001 |
| Hispanic | 0.98 | 0.83 1.15 | 0.812 | 1.09 | 1.02 1.18 | 0.02 | 0.98 | 0.92 1.04 | 0.47 |
| Asian/Pacific Islander | 1.31 | 0.79 2.17 | 0.280 | 1.19 | 1.01 1.41 | 0.04 | 1.13 | 0.98 1.31 | 0.10 |
| Native American | 0.60 | 0.38 0.94 | 0.03 | 0.98 | 0.76 1.25 | 0.84 | 0.92 | 0.75 1.11 | 0.38 |
| Other | 1.30 | 1.02 1.66 | 0.04 | 1.18 | 1.05 1.33 | 0.01 | 1.21 | 1.10 1.33 | < 0.001 |
| Median household income percentile |  | | |  | | |  | | |
| 0–25th | Reference | | |  | | |  | | |
| 26 to 50th | 0.85 | 0.75 0.97 | 0.01 | 0.99 | 0.93 1.05 | 0.67 | 0.96 | 0.91 1.00 | 0.06 |
| 51 to 75th | 0.98 | 0.85 1.12 | 0.73 | 0.95 | 0.90 1.01 | 0.12 | 0.97 | 0.92 1.02 | 0.22 |
| 76 to 100 | 0.79 | 0.67 0.92 | 0.004 | 0.93 | 0.87 1.00 | 0.06 | 0.93 | 0.88 0.99 | 0.02 |
| ACAME[d] | 0.80 | 0.70 0.91 | 0.001 | 0.76 | 0.72 0.80 | <0.001 | 0.73 | 0.70 0.77 | <0.001 |

[a]Multiple imputation and national estimate data.

[b]Short LOS = short length of stay < 2 days.

[c] Long LOS = short length of stay ≥ 2 days.

[d]ACAME = Affordable Care Act Medicaid Expansion.

offer an exhaustive representation of nationwide trends. Notably, the challenge of missing health insurance data in the NTDB, as evidenced by the gap highlighted in Bell et al., presents potential biases [36]. Further, other studies have used datasets from a single site or state, limiting the generalizability of their findings [3,10,14–17]. In contrast, the NIS allows for broader generalizability, capturing a large spectrum of hospital discharges from community-based hospitals across the United States.

This study reveals a complicated relationship between race, LOS and in-hospital mortality which differs from previous reports in the literature [6,7,37–39]. Black patients demonstrate a lower overall in-hospital mortality compared to their White counterparts. However, this trend shifts with LOS. For Black patients experiencing shorter hospital stays, there is a marked increase of in-hospital mortality rates, contrasting sharply with the reference group.

This divergence from existing literature necessitates a closer examination of potential contributing factors. Socio-economic conditions, accessibility to healthcare resources, and disparities in the quality of treatment received emerge as plausible explanations. It is possible that Black patients, particularly those with shorter hospital stays, could have faced systemic barriers leading to delayed or inadequate healthcare, subsequently increasing in-hospital mortality risks. Disparities, especially in prehospital care among Black patients, could result in disparate outcomes [40,41]. Conversely, the prolonged hospitalization of Black patients might facilitate better management of their health conditions, resulting in improved outcomes.

Hispanics and Asian/Pacific Islanders exhibited an increase of in-hospital mortality with longer stays, highlighting a different interaction with the healthcare system. This discrepancy could highlight unique challenges faced by these racial groups, potentially stemming from cultural, linguistic, or systemic biases within healthcare practices [42].

This study's use of administrative databases poses certain limitations—primarily, the potential lack of comprehensive clinical details is a challenge acknowledged in prior research [43–45]. While we employed the 3M™ APR-DRG system, which has been validated for its alignment with trauma registry data particularly in patients requiring intensive care, to mitigate this shortfall, it is important to recognize that this may not completely overcome the limitations inherent to administrative data [44]. Furthermore, it is inappropriate to draw conclusions regarding the impact of ACAME on overall in-hospital trauma mortality, given the staggered adoption of this policy across US states [46].

Future research should consider integrating more detailed clinical data, perhaps by merging administrative databases with clinical registries or by incorporating electronic health record information. Additionally, the lack of data on pre-hospital care and long-term outcomes in this dataset restricted the study's ability to comprehensively assess the long-term impact of insurance status and race on trauma outcomes.

## Conclusion

This study illuminates the intricate relationships between insurance coverage, race, duration of hospitalization, and in-hospital mortality of trauma patients at EROM, underscoring the need for strategic interventions and policy reforms. The findings highlight the necessity of targeted efforts to address the high in-hospital mortality regardless of insurance status during early hospitalization, and the heightened vulnerability of Black patients in shorter hospitalizations. Addressing these disparities requires a systemic approach, ensuring equitable healthcare access and treatment quality across all racial groups, and highlighting the critical role of comprehensive data analysis in informing such efforts.

## Supporting information

**S1 Checklist. STROBE statement—Checklist of items that should be included in reports of observational studies.**
(DOCX)

**S1 Appendix.**
(DOCX)

## Author Contributions

**Conceptualization:** José A. Acosta.

**Formal analysis:** José A. Acosta.

**Methodology:** José A. Acosta.

**Writing – original draft:** José A. Acosta.

**Writing – review & editing:** José A. Acosta.

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
