## [Decision Letter · Decision Letter 0]

3 Oct 2023

PONE-D-23-20866The role of race and insurance in trauma patients’ mortality: A cross-sectional analysis based a nationwide samplePLOS ONE

Dear Dr. Acosta,

Thank you for submitting your manuscript to PLOS ONE. After careful consideration, we feel that it has merit but does not fully meet PLOS ONE’s publication criteria as it currently stands. Therefore, we invite you to submit a revised version of the manuscript that addresses the points raised during the review process.

We look forward to receiving your revised manuscript.

Kind regards,

Kamran Baig, MPH, MBBS

Academic Editor

PLOS ONE

Journal Requirements:

**Additional Editor Comments:**

Thank you for submitting the interesting article. However there are some methodological and technical issues which needs major revision. The abstract requires enhanced clarity concerning the study's design and findings. The introduction lacks a robust foundation to underpin the study's objectives and necessitates reinforcement through a comprehensive literature review. Within the methodology section, there is an abundance of repetitive and convoluted information, and crucial elements such as consent and conflict of interest statements are conspicuously absent. Additionally, the explanation of statistical methods lacks the requisite specificity. In the results section, there are critical misinterpretations of data, notably concerning variables related to length of stay and race. Furthermore, the discussion section lacks practical implications and fails to adequately reflect upon the interpretation of the study's findings. These issues warrant careful attention and rectification to ensure the manuscript meets the standards of professional scholarly communication.

Reviewers' comments:

Reviewer's Responses to Questions

**Comments to the Author**

1. Is the manuscript technically sound, and do the data support the conclusions?

Reviewer #1: Yes

Reviewer #2: No

2. Has the statistical analysis been performed appropriately and rigorously? 

Reviewer #1: Yes

Reviewer #2: No

3. Have the authors made all data underlying the findings in their manuscript fully available?

Reviewer #1: Yes

Reviewer #2: Yes

4. Is the manuscript presented in an intelligible fashion and written in standard English?

Reviewer #1: Yes

Reviewer #2: No

5. Review Comments to the Author

Reviewer #1: none

Reviewer #2: REVIEW OF “THE ROLE OF RACE AND INSURANCE IN TRAUMA PATIENTS’ MORTALITY: A CROSS-SECTIONAL ANALYSIS BASED A NATIONWIDE SAMPLE

The study examines the effect of insurance status, race, and length of hospital stay on inpatient mortality. The study has several points of strength and weakness. All the points that need to be reviewed are discussed section by section as well as the specific location of text in the study.

Abstract

The abstract needs to be reviewed for the clarity of the study question, inclusion and exclusion criteria, and interpretation of the study findings. It should also be reviewed for the redundant text especially when it mentions the effect of race on the findings. The study did not use uniform terminologies throughout the paper e.g., inpatient mortality mentioned in the abstract is referred to as in-hospital mortality in the data table 2.

Introduction

This section explains the previous research done on the topic and reveals the gaps that are to be filled in the current study. However, it fails to convince the reader that in the presence of EMTALA, the trauma patient at extreme risk of mortality with no insurance coverage is at a disadvantage. According to the EMTALA (Emergency Medical Treatment and Labor Act), all patients requiring emergency medical treatment have the right to be treated or stabilized regardless of their insurance status.

Method

This section explains the study design, study settings, and the data source. It is a lengthy section with multiple subheadings with each subheading explaining redundant and incoherent information that can be confusing to the reader. E.g., the “study design” section explains the data source, and then in “Settings” the data source is mentioned again.

The subheading “participants” is inappropriate as the study is performed on the secondary source data and the patient information used in the study is already deidentified to protect confidentiality. Consent for using the data source and statement of Conflict of interest for mentioning a commercial product should also be mentioned. In the subheading “bias” there is an unnecessary explanation of the NIS methodology for data collection and its evolution over the years. There is no mention of the version of APR-DRG used.

Quantitative variables:

The section presents redundant, incoherent information that can be reviewed for clarity. the term LOS throughout the paper has been defined multiple times but at each time the definition is non-homogenous and confusing. Here LOS is defined as early and late groups with the early group having been discharged within two days and the late group with LOS greater than 2 days. Later in the paper, there is mention of 2 groups with LOS of 2 days and others with LOS of more than 2 days without mention of the term “early or late”

Statistical method:

This section relies on the use of approximations e.g., (analysis used a combination of multiple imputation and survey data analysis modules in Stata) and does not clearly mention the specific statistical tests done to control for the missing data like e.g. (expectation maximization method), survival treatment assignment bias (cox regression, Kaplan Meier survival analysis, etc.).

Results:

This section has multiple erroneous data interpretations. LOS here refers to 2 days for one group and more than 2 for another group unlike what has been defined earlier. The interpretation of the racial variable for Table 2 is erroneous.

Discussion:

The discussion is detailed but lacks practical implications of the findings and how they can influence future research. There is a big contradiction in the discussion part where it says that the pattern of LOS and mortality in African Americans is similar to Hispanics and other races which is an incorrect interpretation of the analysis. The study refers to several other studies that found racial disparities in health outcomes but failed to include race-specific insurance data for the sample in the analysis. Had race-specific insurance data been analyzed it would have demonstrated the relation between racial disparity in health insurance and subsequent care.

Specific comments:

Line 26: “70,381 discharges” is a confusing term, it would be clearer if it is substituted by the in-hospital mortality

Line 75: the hospital participating to NIS should be reviewed to match the up-to-date information

Line 80: which version of APRDRG is used?

Line 116: The timing of the insurance coverage does not matter in this condition as the patient is covered by EMTALA

Line 129: this statement does not make sense as the study participants are 65 or younger

Line 133/134: Confusing

Line 143/line 154: there should be uniformity in using terminology line 143 says all community hospitals except long-term care facilities while line 154 says excluding govt institutions where whereas NIS data comprised of all us community hospitals except the long-term care and rehabilitation. This sample definition should be consistent with the NIS 2020 definition. https://hcup-us.ahrq.gov/nisoverview.jsp

Line 147: What were the confounding factors and which statistical analysis was performed to control it.

Line 148 what is meant by survey data analysis modules ??? specify the statistical test used for the study data

Line 155: MCAR (missing completely at random) analysis should be done and the p-value should be mentioned in order to satisfy the assumption.

Line 157: Please specifically mention the statistical test performed for data imputation e.g. expectation maximization method.

Line 177: there is an error in the interpretation

LINE 191: Again, there is nonuniformity in defining the LOS, it is mentioned LOS is 2 days for one category and more than 2 days for another category while initially it was mentioned as 2 groups LOS within 2 days and LOS with 2 days

Line 192: again, the LOS is defined in a different manner

Line 219: LOS is mentioned as a short stay (within 2 days) and long stay, unlike earlier where it is mentioned as early and late

Line 231: 26-50th income percentile has the same effect as 76th to 100th income percentile

Line 256: Hispanic and “other” races show the reverse of the pattern shown by African Americans

Decision:

The study should undergo major revision for their methodology, usage, and interpretation of the data. The findings should be presented with clarity and accuracy to draw logical conclusions from the study question. The terminologies used in the study are inconsistent and should be standardized. It seems that the author is completely unaware of the EMTALA act according to which all patients coming to ER for emergency treatment will be treated immediately and equally regardless of the insurance status. The inference drawn from this study can be misleading to the reader. The mortality may just be related to the severity of the initial trauma. APR-DRG system provides data for the severity of illness, which if considered in the analysis can give a better estimation of the outcome for such patients.

6. PLOS authors have the option to publish the peer review history of their article (what does this mean?). If published, this will include your full peer review and any attached files.

Reviewer #1: No

Reviewer #2: No

---

## [Author Response · Author response to Decision Letter 0]

6 Nov 2023

Dear Academic Editor and Reviewers,

I express my deepest gratitude for the extensive and meticulous review of my manuscript, along with the invaluable feedback that has been provided. Your observations and detailed critiques were invaluable.

The abstract has undergone extensive revision for enhanced clarity, precise definition of the study question, and clear articulation of inclusion and exclusion criteria. Redundant text, especially concerning the effects of race, has been removed. Priority was given to standardizing the terminology across all sections of the paper (e.g., inpatient mortality vs. in-hospital mortality).

I expanded the literature review, addressing the relationship between insurance status, race, and outcomes in trauma patients, and I elaborated on the varied interpretations and conclusions drawn in previous studies. Additionally, I incorporated a discussion on the Emergency Medical Treatment and Labor Act (EMTALA), examining its implications on this study and its role in shaping patient outcomes. 

I revised the entire methods section for brevity, when necessary. I elaborated on the consent exemption process, as well as any potential conflicts of interest. Regarding statistical analysis, I now provide specific details on the tests used and the rationale behind their selection.

Concerning variables associated with length of stay and race in the results section, I fully recognize the mistakes made in my initial interpretation, and I worked to correct these errors to present a more accurate portrayal of the data.

Lastly, I revised the discussion section to elucidate the practical implications of the findings, creating a stronger link between theory and practical application. I also note opportunities for future research.

Specific comments:

Line 26: “70,381 discharges” is a confusing term, it would be clearer if it is substituted by the in-hospital mortality

Thank you for this comment. I understand that the term “discharges” might seem ambiguous, especially when considered within the context of hospital mortality rates. However, as outlined by the National Inpatient Sample (NIS; https://hcup-us.ahrq.gov/db/nation/nis/nischecklist.jsp), the primary unit of analysis in the NIS pertains to inpatient stays rather than individual patients. Hence, “70,381 discharges” denotes the cumulative number of inpatient stays within the sample study, rather than the number of in-hospital mortality incidents. Nonetheless, I revised to “stays” in the text to prevent potential misunderstandings for readers unfamiliar with the NIS terminology.

Line 75: the hospital participating to NIS should be reviewed to match the up-to-date information

Thank you for this comment. This dataset encompasses inpatient discharges from community hospitals in 48 states and Washington DC. By definition, community hospitals include all non-federal, short-term, general, and other specialty hospitals, excluding hospital units of institutions. I amended the manuscript to accurately reflect the current status of hospitals participating in the NIS. 

Line 80: which version of APRDRG is used?

Thank you for this question. I amended the manuscript to reflect that the APRDRG versions were updated annually during the course of the study years.

Line 116: The timing of the insurance coverage does not matter in this condition as the patient is covered by EMTALA

Thank you for pointing out the relevance of EMTALA in the context of emergency care. You are right that EMTALA ensures that every individual is evaluated and stabilized in emergency situations irrespective of their insurance status. However, while EMTALA mandates the provision of emergency care to all, regardless of insurance status, it does not ensure equity in the quality or continuity of subsequent care. Especially for trauma patients at an extreme risk of mortality, the post-stabilization care, which is crucial for their survival and recovery, might be influenced by their insurance coverage and other socio-economic factors. In addition, compliance with the mandate is not necessarily equal across institutions. Nonetheless, including this issue in the manuscript is very relevant, so I revised accordingly.

Line 129: this statement does not make sense as the study participants are 65 or younger

Thank you for bringing this to my attention. You are correct that Medicare typically covers individuals aged 65 years or older. However, it is worth noting that certain younger individuals with disabilities or specific diseases (e.g., end-stage renal disease or ALS) can also be eligible for Medicare benefits. In the NIS dataset, the categorization of insurance type is based on the information provided. While most participants are aged ≥ 65 years, there might be a small subset of younger patients with conditions that make them eligible for Medicare. Nevertheless, I acknowledge that this can cause confusion for the readers. For clarity, I amended the manuscript to reflect the above.

Line 133/134: Confusing

I revised as follows for additional clarity: “Assessing the impact of hospital stay duration on in-hospital mortality involved categorizing duration into two groups based on length of stay (LOS): short LOS, comprising stays less than two days; and long LOS, encompassing stays of two days or more.”

Line 143/line 154: there should be uniformity in using terminology line 143 says all community hospitals except long-term care facilities while line 154 says excluding govt institutions where whereas NIS data comprised of all us community hospitals except the long-term care and rehabilitation. This sample definition should be consistent with the NIS 2020 definition. 

Thank you for highlighting this inconsistency. I ensured the manuscript aligns with the NIS 2020 definition. Specifically, I revised the description to reflect that community hospitals, per the NIS definition, comprise “all non-federal, short-term, general, and other specialty hospitals, excluding hospital units of institutions.” This encompasses not only specialty hospitals, such as those focusing on obstetrics-gynecology, ear-nose-throat, orthopedic, and pediatric care, but also includes public hospitals and academic medical centers.

Line 147: What were the confounding factors and which statistical analysis was performed to control it.

Thank you for your query regarding confounding factors. In this study, I identified age, race, median household income, sex, and length of stay as potential confounders that could influence the relationship between the predictor variable—insurance—and the outcome. To effectively control these potential confounders and more accurately gauge the impact of insurance on the outcome, I employed multivariable regression analyses. In addition, I updated the sensitivity analysis to include an interaction term between insurance and race.

Line 148 what is meant by survey data analysis modules ??? specify the statistical test used for the study data

“Survey data analysis modules” refers to specialized statistical techniques and software functionalities tailored for analyzing complex survey data. Survey datasets, like the one derived from the NIS, have a complex design that includes stratification, clustering, and weighting. In this analysis, I employed Stata’s svy command to account for the complex survey design inherent in the NIS dataset. By using the svy prefix before the statistical analyses, I ensured accurate standard error estimations, which provide more reliable results than traditional methods that do not consider these survey complexities. In the study I employed multivariate logistic regression analyses, adjusted for the NIS’s complex survey design, to analyze the data.

Line 155: MCAR (missing completely at random) analysis should be done and the p-value should be mentioned in order to satisfy the assumption.

I apologize for this oversight. In the original manuscript, the missing data assumption was not MCAR but instead was missing at random (MAR). While the data may not meet the MCAR assumption, it is likely to satisfy the MAR assumption.

Line 157: Please specifically mention the statistical test performed for data imputation e.g. expectation maximization method.

Thank you for this comment. To address missing race data, multiple imputation via chained equations was employed, utilizing multinomial logistic regression analyses to accommodate the categorical nature of the race variable.

Line 177: there is an error in the interpretation

I acknowledge the error and made the necessary changes.

Line 191: Again, there is nonuniformity in defining the LOS, it is mentioned LOS is 2 days for one category and more than 2 days for another category while initially it was mentioned as 2 groups LOS within 2 days and LOS with 2 days & Line 192: again, the LOS is defined in a different manner & Line 219: LOS is mentioned as a short stay (within 2 days) and long stay, unlike earlier where it is mentioned as early and late

I apologize for the inconsistency. I revised to “short LOS” and “long LOS” throughout the manuscript.

Line 231: 26-50th income percentile has the same effect as 76th to 100th income percentile

Thank you. I revised accordingly.

Line 256: Hispanic and “other” races show the reverse of the pattern shown by African Americans

I revised this section to accurately reflect the data, addressing the patterns exhibited by different racial groups. I also made the following revisions for additional clarity:

1) I deleted Tables 2 and 3 from the original manuscript

2) I include an appendix with data from the sensitivity analysis

3) I revised the tables to ensure uniform length of stay data

4) I removed excess subtitles from the methods and results sections

I am dedicated to improving my manuscript to meet the high standards of professional scholarly communication and appreciate the guidance given to help me reach that goal. Please let me know if further revisions are required.

Sincerely,

José A. Acosta MD, MBA, MPH

---

## [Decision Letter · Decision Letter 1]

11 Jan 2024

PONE-D-23-20866R1The role of race and insurance in trauma patients’ mortality: A cross-sectional analysis based on a nationwide samplePLOS ONE

Dear Dr. Acosta,

Thank you for submitting your manuscript to PLOS ONE. After careful consideration, we feel that it has merit but does not fully meet PLOS ONE’s publication criteria as it currently stands. Therefore, we invite you to submit a revised version of the manuscript that addresses the points raised during the review process.

**ACADEMIC EDITOR: **

The authors have submitted a good revision for their manuscript. However, there are outstanding comments from the reviewers which still need to be addressed including: 

"Another limitation that I can see is that you have not stratified the results before and after Affordable Care Act came into force in 2014. This Act improved access to affordable insurance. This stratification could have highlighted any differences in outcomes before 2014 (when less people were insured). Your study had very few uninsured patients."

The authors should address these reviewer comments and once these comments are addressed we would be happy to reconsider the article for publication.

We look forward to receiving your revised manuscript.

Kind regards,

Souparno Mitra, M.D.

Academic Editor

PLOS ONE

Journal Requirements:

Reviewers' comments:

Reviewer's Responses to Questions

**Comments to the Author**

1. If the authors have adequately addressed your comments raised in a previous round of review and you feel that this manuscript is now acceptable for publication, you may indicate that here to bypass the “Comments to the Author” section, enter your conflict of interest statement in the “Confidential to Editor” section, and submit your "Accept" recommendation.

Reviewer #1: All comments have been addressed

Reviewer #3: All comments have been addressed

Reviewer #4: (No Response)

2. Is the manuscript technically sound, and do the data support the conclusions?

Reviewer #1: Yes

Reviewer #3: Yes

Reviewer #4: Yes

3. Has the statistical analysis been performed appropriately and rigorously? 

Reviewer #1: Yes

Reviewer #3: Yes

Reviewer #4: Yes

4. Have the authors made all data underlying the findings in their manuscript fully available?

Reviewer #1: Yes

Reviewer #3: Yes

Reviewer #4: Yes

5. Is the manuscript presented in an intelligible fashion and written in standard English?

Reviewer #1: Yes

Reviewer #3: Yes

Reviewer #4: Yes

6. Review Comments to the Author

Reviewer #1: (No Response)

Reviewer #3: Introduction:

General comment: The Introduction is too brief; it could have elaborated more on the existing data on knowledge gaps on the topic.

Line 68-69: Add reference for the statement "However, while this act guarantees the doorway to immediate medical attention, it stops short of ensuring a uniform quality or continuity of care in the post-stabilization phase."

Methods:

Line 79-80: Please elaborate this statement to make it clearer "The NIS represents 35–39 million annual hospitalizations from 7–8 million raw hospital stays". This will also help in clarifying statements in the Result section (line 126-130) like, "Among these aged 18–64 years, 715,368 inpatient stays met the criteria for trauma, corresponding to a national estimate of 3,512,361. Of these 70,381 had EROM, which correlated with a national estimate of 346,659 stays, which serve as the basis for this study."

Line 107: change "handled" to handle"

Results:

Line 127: Please elaborate the units of 506,403,411 (hospitalizations).

Discussion:

Line 229-238: Another limitation that I can see is that you have not stratified the results before and after Affordable Care Act came into force in 2014. This Act improved access to affordable insurance. This stratification could have highlighted any differences in outcomes before 2014 (when less people were insured). Your study had very few uninsured patients.

Reviewer #4: Dear Dr. Souparno Mitra ,

I have had the honor of reviewing the article titled "The role of race and insurance in trauma patients’ mortality: A cross-sectional analysis based on a nationwide sample” submitted to PLOS ONE for peer review.

The manuscript addresses a very important topic, the relationship between insurance status, race, and in-hospital mortality outcomes for trauma patients at extreme risk of mortality (EROM). The study undertook the exploration of these topics by analyzing data from the National Inpatient Sample (NIS) spanning from 2007 to 2020. The analysis focused on trauma patients aged 18–64 years excluding those with primary burn or non-trauma diagnoses. The study utilized the All Patient Refined Diagnosis Related Groups (APRDRG) system to categorize patients based on severity and risk. Insurance status was the primary exposure investigated concerning in-hospital mortality among trauma patients categorized as Emergency Room (EROM). Insurance classifications included uninsured (self-payers and those with missing data) and insured (Medicare, Medicaid, or private policies). Hospital stay duration was categorized into short (less than two days) and long (two days or more). The study employed a time-varying multivariate logistic regression model, adjusting for age, race, sex, and median household income. The NIS dataset used in the study is fully anonymized to protect patient privacy. The research was classified as non-human subject research by the Solutions IRB, LLC, exempting it from the requirement for informed consent. Statistical analysis was conducted using Stata version 18 MP, with significance set at p < .05.

The author found that demographic-wise, among the 70,381 EROM patients, they were predominantly white men and fell into the lowest income quartile. Descriptive data revealed that 23.7% had a short length of stay (LOS), and 76.3% had a long LOS. In the group in the short LOS, the in-hospital mortality rate stood at 74.3%, whereas it was notably lower at 17.4% in the long LOS group. Insurance coverage was widespread in both cohorts, constituting 74.8% for the short LOS group and 86.2% for the long LOS group. Results from the multivariate logistic regression analysis revealed a 36% lower likelihood of in-hospital mortality for insured patients compared to their uninsured counterparts. However, this impact varied based on the length of stay, exhibiting no significant effect in the short LOS group and a notable 15% reduction in the long LOS group. In comparison to White patients, the authors found that Black patients exhibited a 10% decrease in the odds of in-hospital mortality. Within the short LOS group, Black patients faced an 18% increase in mortality odds, whereas Native Americans experienced a substantial 40% reduction. In the long LOS group, Black patients encountered a 9% decrease, while Hispanics and those classified under the 'other race' category saw a 15% and 20% increase in mortality odds, respectively. Concerning median household income, individuals in the 26–50th percentile witnessed a 10% reduction in in-hospital mortality within the short LOS group, whereas those in the highest income bracket (76–100th percentile) experienced a 15% decrease.

The highlights of the study include the research on having insurance and its relation to mortality rates in patients at EROM. The relationship between insurance status, race, and in-hospital mortality outcomes for trauma patients at extreme risk of mortality (EROM). Trauma is a global public health challenge and is a leading cause of mortality and prolonged disability. To help uninsured patients, a law in 1986 made sure that emergency departments provide care for critical patients but the hospital does not help with post-stabilization care. Furthermore, in past literature, the authors find that Black trauma patients have disproportionately adverse outcomes. The authors’ goal is to raise awareness of healthcare equity and provide insights for informed policy interventions. They suggest conducting further research with more descriptive clinical data like looking into the health record of the patients. They also suggest more research be done in the pre-hospital care and long-term outcomes of having or not having insurance.

This study has a few limitations. The limitations have been accounted for in the methodology of the study. The principle of "missing at random" was employed to manage the missing data, taking into account the nature and data collection methodology of the NIS. Specifically, the race variable had 10.07% missing data, whereas all other variables had less than 5% missing. Multiple imputation via chained equations was utilized to handle the missing race data, employing multinomial logistic regression analysis to accommodate the categorical nature of the race variable. They used NIS which allows for the conclusions to be applied to a larger population of the United States. However, using data from NIS is also a limitation as it is not comprehensive of the patients’ situations.

7. PLOS authors have the option to publish the peer review history of their article (what does this mean?). If published, this will include your full peer review and any attached files.

Reviewer #1: No

Reviewer #3: **Yes: **Amit Jagtiani

Reviewer #4: No

---

## [Author Response · Author response to Decision Letter 1]

14 Jan 2024

Reviewer #3: Introduction:

General comment: The Introduction is too brief; it could have elaborated more on the existing data on knowledge gaps on the topic.

Thank you for your thorough review and constructive feedback on this manuscript. Your suggestion to expand the introduction has been invaluable. In response, I have revised and extended this section to address the existing knowledge gaps more comprehensively regarding the interaction between insurance, race, and in-hospital mortality in trauma patients at extreme risk of mortality.

The revised introduction now includes additional context and references to emphasize the current state of research in this area. These enhancements aim to clarify the specific gaps this study seeks to address and articulate the relevance of this research. The revisions also improve the flow and coherence between paragraphs, ensuring a smoother transition of ideas for the reader. I believe these changes substantially enrich the manuscript and provide a clearer framework for understanding the significance of this study's contributions.

Line 68-69: Add reference for the statement "However, while this act guarantees the doorway to immediate medical attention, it stops short of ensuring a uniform quality or continuity of care in the post-stabilization phase."

References have been added to the above statement in the introduction.

Methods:

Line 79-80: Please elaborate this statement to make it clearer "The NIS represents 35–39 million annual hospitalizations from 7–8 million raw hospital stays". This will also help in clarifying statements in the Result section (line 126-130) like, "Among these aged 18–64 years, 715,368 inpatient stays met the criteria for trauma, corresponding to a national estimate of 3,512,361. Of these 70,381 had EROM, which correlated with a national estimate of 346,659 stays, which serve as the basis for this study."

I appreciate the reviewers highlighting an area of the manuscript which lacks clarity. The National Inpatient Sample (NIS) is a large publicly available database which uses a complex survey design allowing for the creation of national estimates. It is notable for its use of weighted estimates. Each record in the database comes with a discharge weight, compensating for the sampling design and the probability of hospital selection. Applying these weights in statistical analysis allows researchers to extrapolate findings to the national level, making the results broadly applicable to most of the U.S. hospital discharge population. 

The data reported in the NIS is based on hospitalizations and not unique individuals. In other words, individual observations are hospitalizations events rather than unique patients [1]. In order to account for this and improve readability the following statement was added to the manuscript. “In the NIS, each record represents a hospitalization event, not a unique patient [1]. For this study, to simplify statistical analysis and interpretation, each hospitalization is considered and referred to as a separate patient encounter. This approach is applied even in cases where individuals may have more than one admission for the same clinical issue.” The manuscript has been revised to consistently refer to 'patients' instead of 'hospitalizations' for improved clarity and coherence.

Line 107: change "handled" to handle"

Thank you for pointing this out. The manuscript has been corrected. 

Results:

Line 127: Please elaborate the units of 506,403,411 (hospitalizations).

Thank you for this inquiry. The manuscript has been revised to consistently refer to 'patients' instead of 'hospitalizations' for improved clarity and coherence. Please see above in the response to the Methods section comments for more detailed explanation.

Discussion:

Line 229-238: Another limitation that I can see is that you have not stratified the results before and after Affordable Care Act came into force in 2014. This Act improved access to affordable insurance. This stratification could have highlighted any differences in outcomes before 2014 (when less people were insured). Your study had very few uninsured patients.

Thank you for the valuable suggestion regarding the stratification of results in relation to the Affordable Care Act Medicaid Expansion (ACAME) in 2014. In response, I have incorporated an ACAME covariate into the multivariate logistic regression model and updated the results accordingly. These changes are now reflected in Table 2 and have been discussed in the revised manuscript. I am grateful for this insightful recommendation.

Reviewer #4: 

I am grateful for the comprehensive review and insightful feedback provided reviewer #4. The reviewer’s acknowledgment of the significance of this research topic is greatly appreciated.

---

## [Decision Letter · Decision Letter 2]

1 Feb 2024

The role of race and insurance in trauma patients’ mortality: A cross-sectional analysis based on a nationwide sample

PONE-D-23-20866R2

Dear Dr. Acosta,

We’re pleased to inform you that your manuscript has been judged scientifically suitable for publication and will be formally accepted for publication once it meets all outstanding technical requirements.

Kind regards,

Souparno Mitra, M.D.

Academic Editor

PLOS ONE

Additional Editor Comments (optional):

Reviewers' comments:

Reviewer's Responses to Questions

**Comments to the Author**

1. If the authors have adequately addressed your comments raised in a previous round of review and you feel that this manuscript is now acceptable for publication, you may indicate that here to bypass the “Comments to the Author” section, enter your conflict of interest statement in the “Confidential to Editor” section, and submit your "Accept" recommendation.

Reviewer #4: All comments have been addressed

Reviewer #5: All comments have been addressed

2. Is the manuscript technically sound, and do the data support the conclusions?

Reviewer #4: Yes

Reviewer #5: Yes

3. Has the statistical analysis been performed appropriately and rigorously? 

Reviewer #4: Yes

Reviewer #5: I Don't Know

4. Have the authors made all data underlying the findings in their manuscript fully available?

Reviewer #4: Yes

Reviewer #5: Yes

5. Is the manuscript presented in an intelligible fashion and written in standard English?

Reviewer #4: Yes

Reviewer #5: Yes

6. Review Comments to the Author

Reviewer #4: Thank you for addressing the comments for the manuscript titled " The role of race and insurance in trauma patients’ mortality: A cross-sectional analysis based on a nationwide sample" submitted to PLOS ONE for peer review.

All the comments were appropriately addressed, and corresponding changes reflect the reviewers’ recommendations.

Reviewer #5: Manuscript is much better after two revisions. All comments seems addressed. I wish good luck to the author.

7. PLOS authors have the option to publish the peer review history of their article (what does this mean?). If published, this will include your full peer review and any attached files.

Reviewer #4: No

Reviewer #5: No

---

## [Editor Report · Acceptance letter]

5 Feb 2024

PONE-D-23-20866R2 

PLOS ONE

Dear Dr. Acosta, 

I'm pleased to inform you that your manuscript has been deemed suitable for publication in PLOS ONE. Congratulations! Your manuscript is now being handed over to our production team.

Kind regards, 

on behalf of

Dr. Souparno Mitra 

Academic Editor

PLOS ONE